# Winter Orographic Precipitation and ENSO in Sapporo, Japan

**Akiyo Yatagai [1],* and Chisato Kinoshita [2]**

[1]  Graduate School of Science and Technology, Hirosaki University, 3 Bunkyocho,
   Hirosaki 036-8561, Aomori, Japan
[2]  Faculty of Science and Technology, Hirosaki University, 3 Bunkyocho, Hirosaki 036-8561, Aomori, Japan
*  Correspondence: yatagai@hirosaki-u.ac.jp; Tel.: +81-172-39-3685

**Abstract:** The effect of global climate change on the distribution of snow water is a great concern. Thus, it is important to clarify the characteristics of winter precipitation variability, including mountain precipitation, together with climate indices. In this study, regional snowfall characteristics were investigated with the daily gridded precipitation over Sapporo City (located on the Japan Sea side of Hokkaido in northern Japan), which was quantified by the APHRODITE method and by adding local precipitation observation data. We found places of showing large interannual variability that is different from that of daily precipitation variability. Applying an EOF analysis to the daily grid precipitation, we defined four local precipitation types. The occurrence of each precipitation type and associated atmospheric circulation was analyzed, and the results revealed that (except for the Super El Niño winter of 1997/1998) more snow fell in the southwestern mountains and inland areas during El Niño winters, and more snow fell in the northeastern plains and along the sea during La Niña winters. Continued development and evaluation of the precise data that incorporate local precipitation network is needed.

**Keywords:** precipitation; snowfall; Hokkaido; APHRODITE; El Niño; La Niña; ENSO

## 1. Introduction

There are growing concerns that global warming increases the amount of water vapor in the atmosphere, which will increase the amount of precipitation [1]. In addition, local environmental changes due to global warming have been studied in global climate model experiments, e.g., the Climate Model Intercomparison Project [2]; however, these climate model investigations have not reproduced the local precipitation distribution corresponding to the global signal, climate index, and teleconnection (e.g., [3]). Thus, it is important to investigate the relationship between the global signal and local precipitation after quantitative evaluation in terms of seasonal prediction. It contributes understanding and forecasting future mountain snow and ice water resources [4], and adaptation to global warming and climate change.

The Honshu Island of Japan Sea side of Japan (Figure 1) is known worldwide for its heavy snowfall due to the winter monsoon (Figure 2a). Although there have long been studies on mountain/plain snow along the coast [5,6], synoptic scale weather impact to them [6,7], Japan convergence zone (JPCZ) [8], and mountain effects to local precipitation [9], there are few studies on long-term variations of snowfall, including those in Hokkaido. One reason for less studies may be the difficulty of snowfall observation.

Since several decades ago, studies of global- or continental-scale climate signals and regional precipitation have been conducted at various spatial and temporal scales; however, the study of the relationship between water resources and the climate index has been limited due to limited precipitation data for mountainous regions. Thus, we collected as much rain-gauge data as possible to create a quantitative grid precipitation dataset over Asia including Japan [10,11]. As these APHRODITE papers show, better climatology can be used for high-resolution model validation including mountainous regions such as the Himalayas [12].

Furthermore, by collecting and including daily precipitation data, these gridded data made it possible to meteorological/monsoonal analysis over the mountainous regions (Yatagai et al., 2012 [11], Figure 7). On the other hand, although the Japan Meteorological Agency (JMA) AMeDAS is a dense-network of rain gauges (1300 stations~), the additional data other than JMA stations have enabled a more quantitative study of the disastrous heavy precipitation including mountainous regions [13].

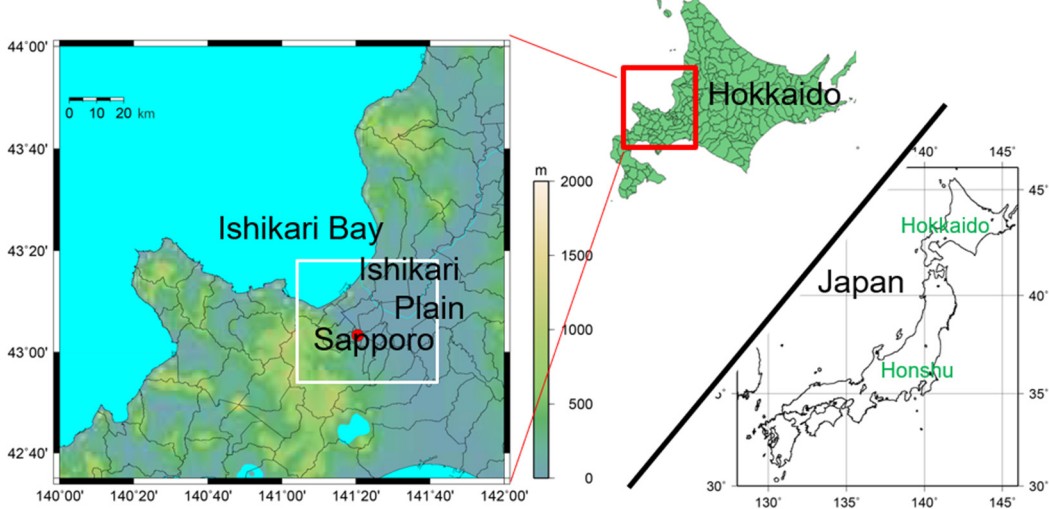

**Figure 1.** Geographic distribution of Hokkaido and area of interest. A red rectangle indicates the domain of the left panel, and a white rectangle indicates the target area shown in Figures 2b and 3. (The left panel was created by processing an electronic topographic map by the Geospatial Information Authority of Japan).

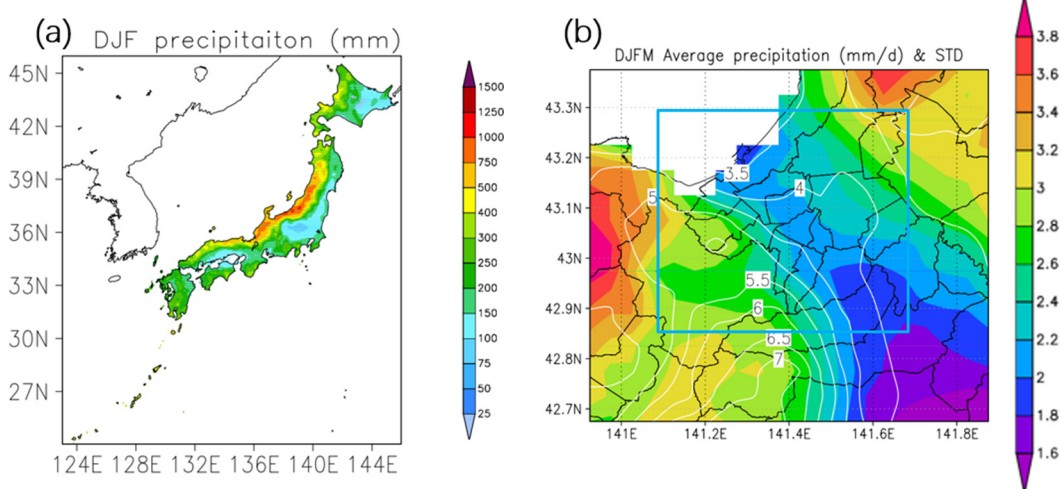

**Figure 2.** (**a**) Averaged winter (December, January and February) precipitation (unit: mm/3 months) for 1961–2008 by APHRO_JP [10]; (**b**) daily averaged precipitation (unit: mm/day, shown in color shade) and standard deviation (white contours) used in this study (December 1993–March 2015, December, January, February, and March were used). A light-blue rectangle indicates the target area.

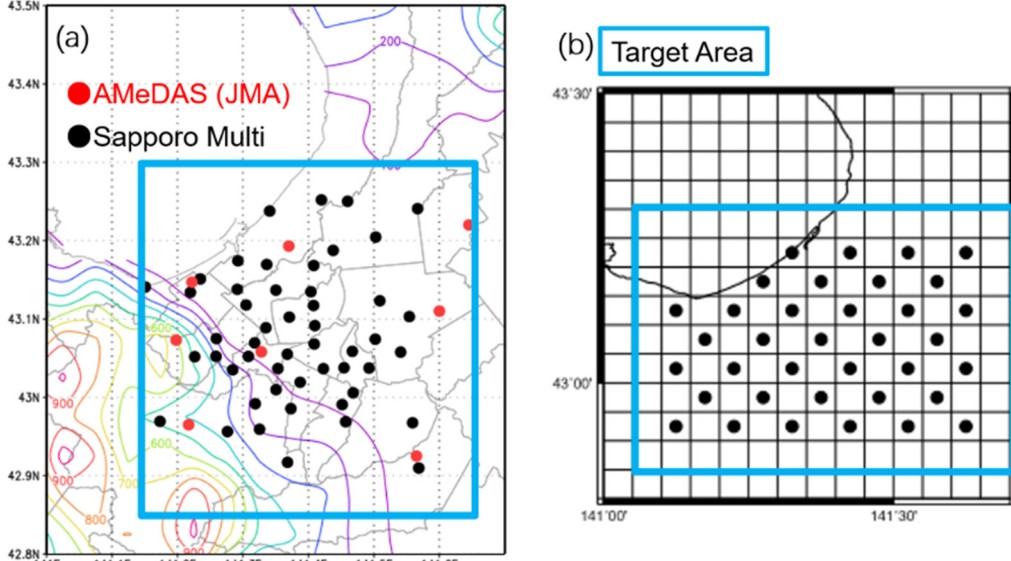

**Figure 3.** (**a**) Distribution of precipitation station by AMeDAS (red) and Sapporo Multi (black) within the target area (a light-blue rectangle). The colored contours in the left panel indicate elevation (orography); (**b**) daily precipitation at the 36-grid points in the target area (a light-blue rectangle) was the subject of EOF analysis in this study.

The Sapporo City region, which is located on the Sea of Japan side of Hokkaido, experiences heavy snowfall. In this region, plains extend from north to east, and mountains are found to the west (Figure 1). Sapporo is located in the northwest area of the Ishikari Plain (see Figure 1). In terms of winter precipitation amount, Sapporo has less than the other cities on the Japan Sea side of Honshu, but with a population of about 1.9 million (2022), Sapporo can be a considered the largest city on the Sea of Japan side. Therefore, the distribution of snow in winter has a significant impact on the lives of citizens [14,15]. In this regard, Sapporo city has its own precipitation observations for road snow removal purposes (Figure 3a). By adding these observations, it would be possible to produce a correspondence with synoptic and large-scale conditions to determine which areas receive more snowfall as a winter season prediction. Figure 2b shows the average daily precipitation for Sapporo during December–March. This is the average precipitation amount obtained through this study. Precipitation is heavy in the mountainous areas distributed in the southwest, and is also heavy in the northeast. The largest standard deviations (SD) of daily precipitation variability are found in the southwestern mountainous region. Using this network data is expected to contribute to reveal regional snowfall patterns and its interannual variability.

Therefore, this study attempts to clarify the regional characteristics of precipitation distribution over Sapporo by adding the local (Sapporo city) precipitation data to AMeDAS stations and creating daily grid precipitation data that expresses orographic effects. Then, use the data to investigate the relationship between local precipitation distribution and synoptic weather, and their interannual variations in winter.

As a precipitation variation over Sapporo, Tachibana (1995) [16] investigated the synoptic and meso-scale statistics of precipitation around Sapporo by applying rotated empirical orthogonal function analysis of precipitation over western Hokkaido. Tachibana explained that snowfall along the coast of Ishikari Plain (see Figure 1) is due to the convergence of katabatic winds on the Hokkaido scale and northwesterly monsoon.

Recently, Farukh and Yamada [15] discussed synoptic-scale atmospheric circulation patterns that are associated with extreme heavy snowfall in Sapporo City. Heavy snowfalls frequently cause traffic problems in Sapporo City. In fact, they have shown the topmost snowfall events were triggered by the advection of very cold airmass from eastern Siberia, anomalously huge moisture with northerly strong wind, active and stationary Aleutian

low, and 500 hPa deep cold-core low over the southern Hokkaido. However, sub-regional (i.e., within Sapporo City) precipitation distribution has not been discussed yet.

Although Japan's winter season is considered to be warm with light snow in the El Niño case [17,18], the relationship with the global signal is expected to be complex [3] due to the effects of winter monsoon intensity, water vapor transport processes, local mountain and valley winds, and sea-land winds. Hence, Sapporo is considered to be one of the best test fields to investigate the impact of El Niño/Southern Oscillation (ENSO) to local snowfall, if quantitative grid precipitation data are created.

The goal of this study was to understand how global signals, e.g., El Niño and La Niña, affect the regional distribution of snowfall in the Sapporo region. However, to do this, reliable data are necessary. Hence, the purpose of the study is (1) to make a daily gridded precipitation dataset by the APHRO_JP method by adding Sapporo Multi-Sensor network data, (2) to analyze regional differences in winter daily precipitation around Sapporo to determine the occurrence of characteristic heavy snow patterns and (3) investigate the relationship of (2) with ENSO.

## 2. Data and Methods

### 2.1. Data Used for Analysis

2.1.1. Precipitation Data

In addition to AMeDAS observation data measured by the Japan Meteorological Agency (JMA), precipitation data by the Sapporo Multi-Sensor Network (hereafter MSS), a meteorological observation network across 57 locations in Sapporo city for snow removal from roads, were used (Figure 3a). The accumulation time of a day is Japan Standard Time (15 UTC-15 UTC) for both AMeDAS and MSS. Here, the analysis period was January 1992 to March 2015. In this study, the four months from December 1992 to March 1993 were defined as the 1993 winter season, and the 23 winter seasons from 1993 to 2015 were considered the period of interest.

2.1.2. Gridding Method and Gridded Data

To represent orographic precipitation, "—more precipitation by orographic lifting of moist air—", we analyzed daily station data on a $0.05° \times 0.05°$ grid using the APHRO_JP method [10]. The APHRODITE algorithm [10,11], i.e., the method of interpolating daily ratio to daily climatology, is useful not only for representing orographic precipitation, but also for complementing missing values and/or sparse data distribution densities.

The APHRO_JP data [10] are comprised of more than 100 years of data. We used an additional 57 stations of MSS data for this study. The comparison of the winter (December–March) precipitation of APHRO_JP and MSS for the same period is shown in Figure 4a,b. MSS shows slightly smaller winter precipitation in the target region. The black contours in Figure 4a,b are a standard deviation of interannual variability of winter precipitation by APHRO_JP and MSS, respectively. Interestingly, in the target area, where additional MSS data used have larger standard deviation (SD), and the SD peak appears along the mountain foot (southwestern part of the target area) and the northeastern part of the target area (141.5° E, 43.15° N). It is worth noting that the location of these peaks of SD in the interannual variability (Figure 4b) is different from the same in daily variability (Figure 4d, which is identical with Figure 2b). For reference, Figure 4c shows the mean and SD of APHRO_JP's 55-year (1961–2016) winter precipitation. This is almost the same as Figure 4a.

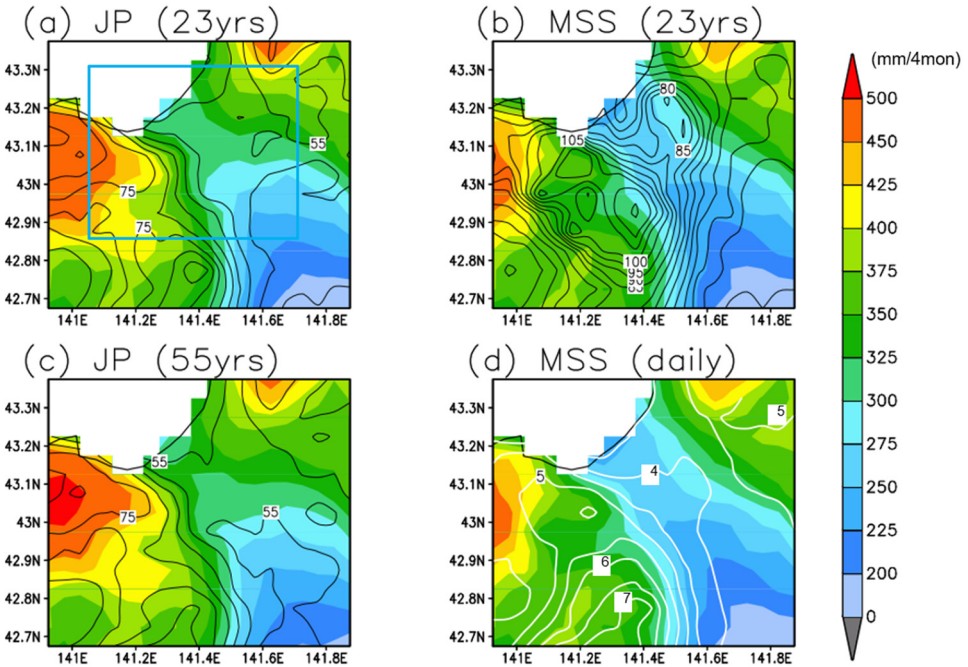

**Figure 4.** (**a**) Seasonal (December–March) averaged precipitation (color) for 1993–2015 by APHRO_JP. Black contours indicate standard deviation of seasonal precipitation during the 23 years (1993–2015). A light-blue rectangle indicates the target area; (**b**) same as (**a**) but for APHRO_JP + Multi Sensor Sapporo network (MSS); (**c**) same as (**a**) but for the 55 years (1961–2016); (**d**) same as Figure 2b (daily averaged precipitation and standard deviation. Unit of the precipitation (color) is mm per the four months (December–March). White contours indicate standard deviation (unit: mm/day, the same with that of Figure 2b).

### 2.1.3. EOF Analysis

From the data collected over the 23 winter seasons, we selected days where more than two-thirds of the stations (Figure 3a) observed precipitation greater than 0.1 mm/day, in order to exclude an almost no precipitation day that would be inappropriate for performing EOF analysis. As a result, 1789 days were subjected for the EOF analysis.

The daily precipitation grid was created as a $0.05° \times 0.05°$ grid (approximately the length of a grid cell is 5 km), and the EOF analysis was performed using the 36 grids shown in Figure 3b as variables (1789 samples). We computed a correlation coefficient matrix of the 36 variables, and then we obtained 36 eigenvectors.

The variance of EOF1 accounted for 85.0% of total variance, while EOF2 and EOF3 accounted for 7.6% and 4.1%, respectively. Figure 5 shows the horizontal distribution of the first three eigenvectors. The correlation coefficients between the EOF score and average precipitation of the target area was $r = 0.99$ (Figure 6); thus, EOF1 represents precipitation over the entire target area, and EOF2 and EOF3 extract modes that represent regional characteristics. In more detail, the center of EOF1 locates in the southern part of the target area, which corresponds to the area that shows a large standard deviation of daily precipitation variability (Figure 2b). The white marks "O", "A", "B", "C", and "D" indicate the locations of peak eigenvectors to easily identify each point in the later diagrams and description.

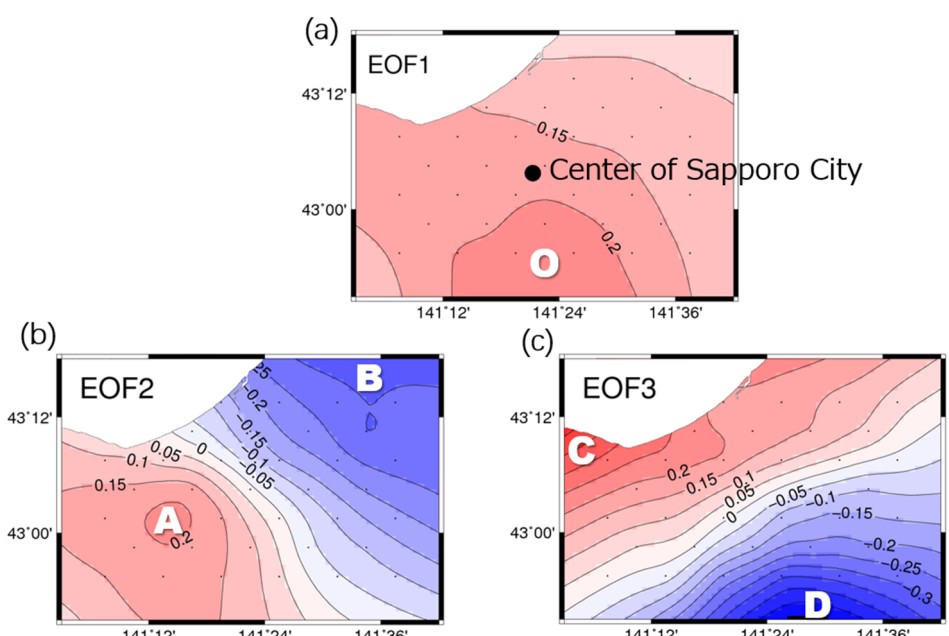

**Figure 5.** Horizontal distribution of eigenvectors of (**a**) EOF1, (**b**) EOF2, and (**c**) EOF3. The contour interval is 0.05, and red/blue indicates positive/negative. The dots in each map indicate the 36 grids subjected to EOF analysis. The mark "O" indicates the maximum of the eigenvector of EOF1, the marks "A" ("B") indicate the maxima of positive (negative) eigenvector of EOF2, and "C" ("D") indicate the same for EOF3.

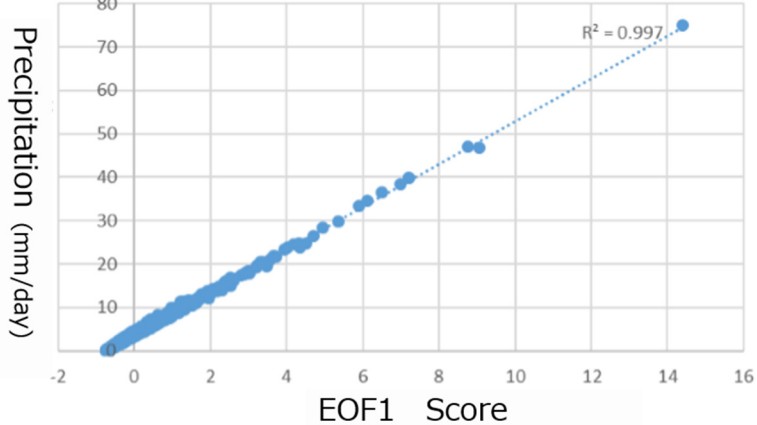

**Figure 6.** Scatter diagram between EOF1 score and daily averaged precipitation of the 36 grid points (Figure 3b). The dotted line indicates a regression line, and the determination coefficient is 0.997.

### 2.1.4. Meteorological Reanalysis Data

We used sea level pressure (SLP) and horizontal wind at 850 hPa from the ERA-interim reanalysis [19], which were assembled by the European Centre for Medium-range Weather Forecasts (ECMWF). The horizontal resolution is 0.75 degree grid, and we computed daily mean wind field from the four times daily product so that it represents the same day with the precipitation data (i.e., averaged 18 UTC data of the previous day, 00 UTC, 06 UTC and 12 UTC of the day).

### 2.1.5. Sea Surface Temperature (SST) Data

Nino3-Sea Surface Temperature (SST) from the JMA (https://www.data.jma.go.jp/gmd/cpd/db/elnino/index/dattab.html, accessed on 1 March 2022) was used as an indicator of ENSO. The reason for using Nino-3 SST is that JMA uses Nino-3 SST for the identification of El Niño and La Niña periods. The JMA defines an El Niño (La Niña) as the

five-month moving average of the difference from the reference value of SST in the monitoring area, being +0.5 °C or higher (−0.5 °C or lower) for six or more consecutive months. (https://www.data.jma.go.jp/cpd/data/elnino/learning/faq/elnino_table.html, accessed on 1 March 2022).

## 3. Results

### 3.1. EOF Results

The geographical characteristics of EOF2 and 3 (Figure 5b,c) are as follows: EOF2 (+) represents mountain (southwest ward) precipitation, EOF2 (-) represents plain precipitation type (northeast ward), EOF3 (+) represents seaward (coastal) precipitation (northwest), and EOF3 (-) represents inland precipitation (south). Based on this precipitating area, we named the four types as mountain precipitation type (A), plain precipitation type (B), seaward precipitation type (C), and inland precipitation type (D). To confirm the precipitation patterns, the composite (average) daily precipitation of the target area for cases A, B, C, and D with the days of absolute value of the score of each EOF is 3.5 or higher, which is shown in Figure 7. These panels (Figure 7) reflect the characteristics of the regional precipitation, i.e., (A) mountain, (B) plain, (C) seaward, and (D) inland, and location can be identified with the positive/negative peak of eigenvectors of EOF2 and EOF3 (Figure 5b,c). Interestingly, these marked areas showed high standard deviation in interannual variability (Figure 4b).

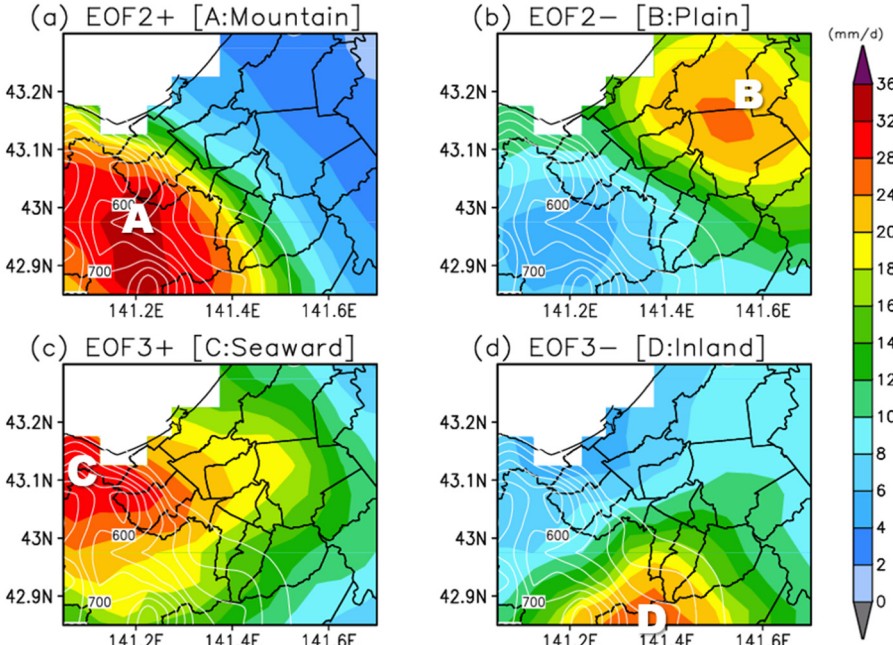

**Figure 7.** Distribution of precipitation composites for each type (unit: mm/day). (**a**) mountain, (**b**) plain, (**c**) seaward, and (**d**) inland types. White contours indicate orography, and black lines indicate administrative boundaries. The marks "A", "B", "C", and "D" are the locations based on Figure 6.

Since EOF1 represents the variability of total precipitation in the target area, discussing EOF2 and 3 is to look at regional differences. However, since the variance of EOF1 is very high (85%) compared with that of EOF2 and 3, the relationship of EOF2 and 3 with EOF1 should be noted and will be discussed later (Section 3.4). We will only point out here that, as shown in Figure 6, there were also many small negative and positive values in EOF1, and that EOF1 scores more than SD = 1 (EOF1 > 1) were found in 184 cases among the 1789 cases that were subjected to EOF analysis.

### 3.2. Wind Direction

The composited 850-hPa wind patterns for the four types (i.e., mountain, plain, seaward, and inland) are shown in Figure 8. The wind systems for the four typical precipitation types are quite different from each other, e.g., A is north wind, B is west wind, C is westnorthwest wind, and D is southwest wind. The wind pattern of B and C reflects westerly high and easterly low pressure (i.e., a common winter pattern), and the composited sea level pressure pattern (not shown) can be identified with the winter pattern, while A and D demonstrated a weak winter pattern and were influenced by migratory low and/or south coast low pressure patterns.

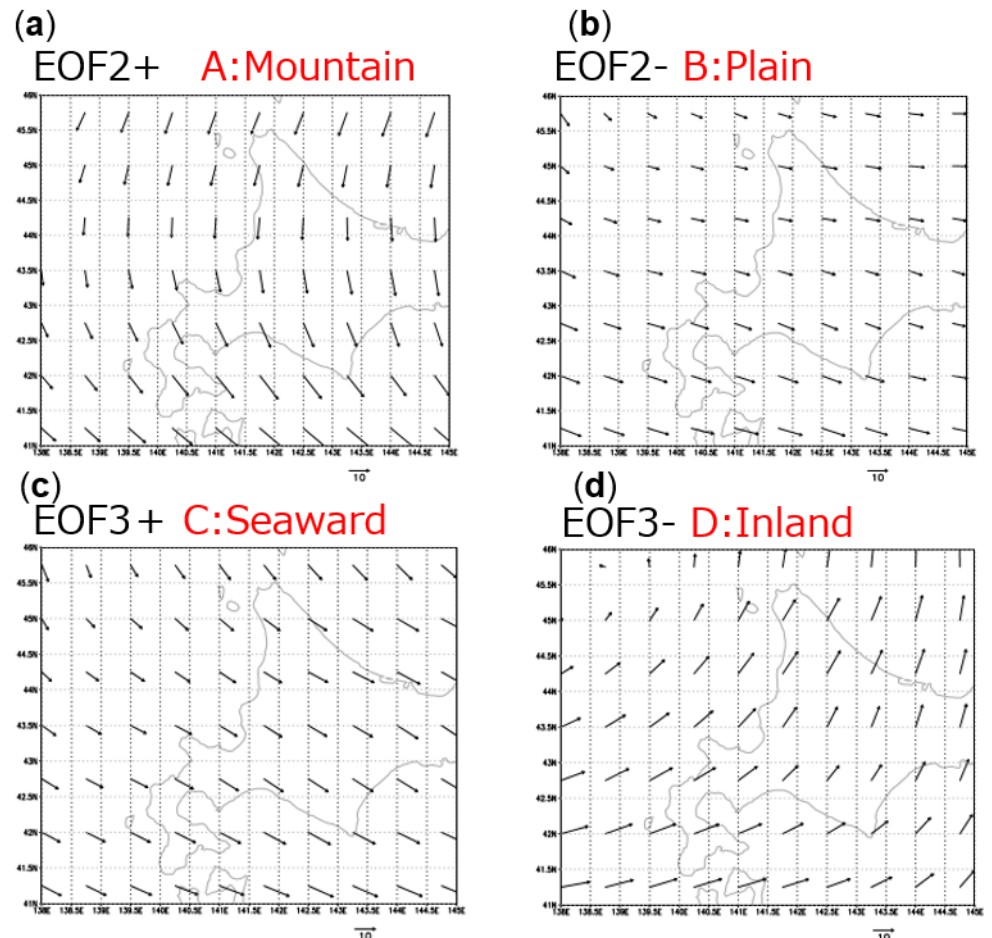

**Figure 8.** Distribution of wind vectors at 850 hPa level. (**a**) mountain, (**b**) plain, (**c**) seaward, and (**d**) inland patterns.

### 3.3. Occurrence of Each Pattern and Its Interannual Variation

As described in Section 3.2, patterns B and C are closely related to winter patterns, and patterns A and D tend to appear when winter patterns are weak. Then, the number of days when the absolute score was greater than 1 was counted from the normalized score time series. The results are summarized in Figure 9. Based on the given criterion, we found that patterns A, B, C, and D appeared in the 23-winter period 166, 207, 176, and 126 times, respectively. Figure 9 demonstrates that patterns B and C are most likely to appear coherently, and A and D are most likely to appear coherently. Furthermore, patterns B and C appear more frequently in La Niña years.

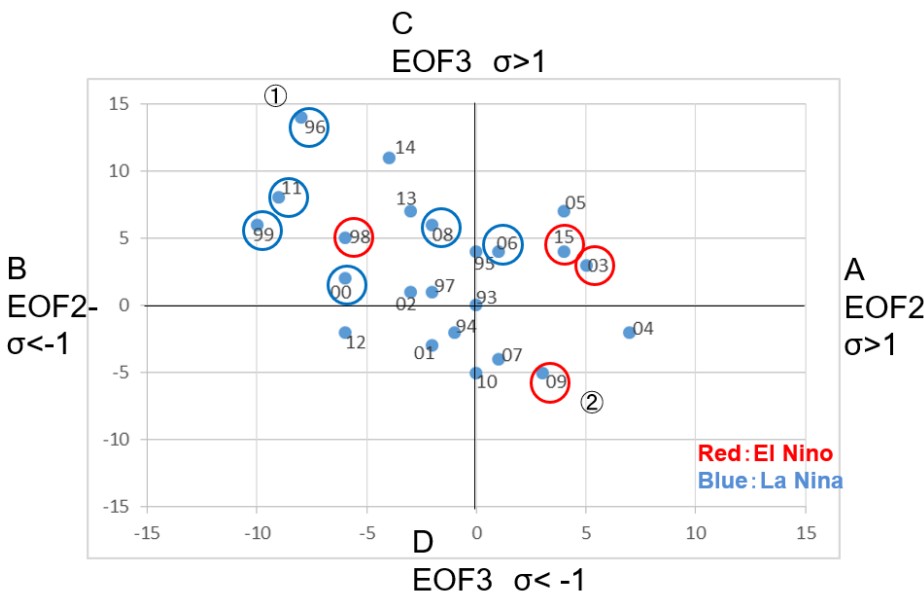

**Figure 9.** Frequency of each mode (number of days on which the absolute score exceeds 1). Numbers indicate year (e.g., 98 means the 1997/1998 winter). Red/blue circles indicate El Niño/La Niña winters.

Figure 10 summarizes the amount of precipitation and occurrence frequency in each precipitation type. The red dots indicate the frequency of EOF1 > 1 (184 cases) for each winter. In years with high precipitation, the frequency of EOF1 > 1 is also high (10 days or more). Years with low EOF1 > 1 days and high precipitation tend to have high frequency of type-B (plain type, 1997, 2011, 2012 and 2014). The correlation with the areal precipitation and EOFs (absolute score greater than 1) is positive for A–D, with B having the highest correlation among the four patterns.

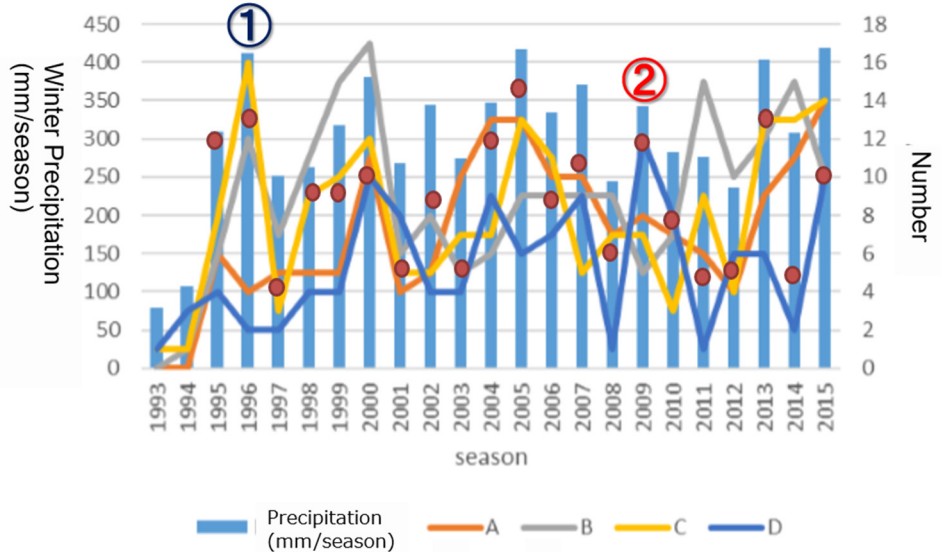

**Figure 10.** Interannual variation of winter precipitation in target region and the number (day) of appearance of each pattern (where the absolute score exceeds 1). Red dots indicate the number (day) of appearance of EOF1 where the score exceeds 1. ① and ② indicate the year (winter) whose composite sea level pressure (SLP) patterns are shown in Figure 11.

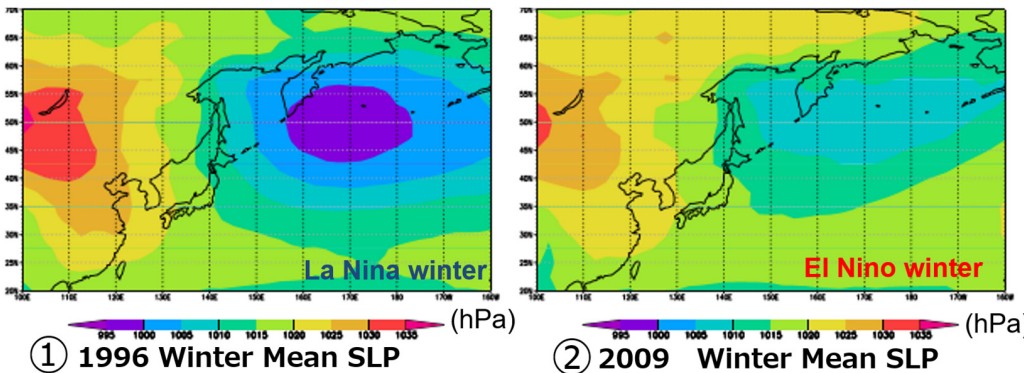

**Figure 11.** Winter mean sea level pressure (SLP) for years (**left**) 1996 and (**right**) 2009.

As shown in Figure 10, the total precipitation is higher in both 1996 (La Niña) and 2006 (El Niño); however, a comparison of circulation fields is attempted for two contrasting years with higher precipitation and EOF1 > 1; however, a clear difference can be observed, i.e., ① having more BC and less AD, and vice versa.

We show SLP of the two winters (Figure 11). Figure 11 clearly shows that the year 1996 (La Niña) exhibited a stronger winter pattern on average. This is consistent with the characteristics of ENSO's East Asian influence summarized by Wang et al. [17], and is also consistent with B and C (Figure 8), which have strong westerly and northwesterly winds when the winter pattern is strong. Let us compare the appearance of B and C with those of A and D.

The difference between the number of days of occurrence for patterns B and C and that for patterns A and D in each year (S = B + C − A − D) is shown in Figure 12 together with the interannual variation of Nino3-SST. With some exceptions (i.e., 1998 and 2009), we found that S and Nino3-SST are negatively correlated, i.e., the correlation coefficient for the 23 winter months is R = −0.27. By excluding 1998, the coefficient is R = −0.58, which is a significant correlation at the 99% confidence level. This indicates that, except for 1998, which is a very strong El Niño, when the SST in the eastern equatorial Pacific is high (El Niño), precipitation is more likely to occur in the mountain and inland regions. In addition, when the SST in the eastern equatorial Pacific is low (La Niña), precipitation is more likely to occur in the plains and oceanic regions.

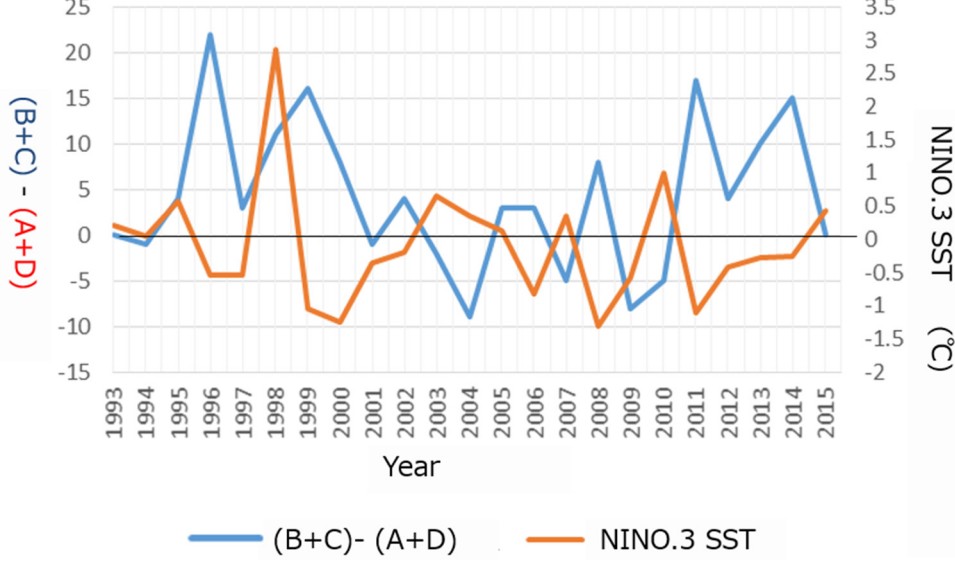

**Figure 12.** Relationship between the number of pattern appearance difference (B + C) − (A + D), and Nino.3 SST. Numbers on the *x*−axis indicate year (e.g., 98 means the 1997/1998 winter).

### 3.4. Occurrence of Type A–D and EOF1

Before moving on to the El Niño/La Niña precipitation composite, we will examine the contribution of EOF 1, which had a large contribution, and types A, B, C, D, which are regional precipitation indicators. Table 1 shows the number of days with absolute scores > 1 for A,B,C,D when EOF1 > 1, and the same for EOF1 < 1. The number of occurrences of El Niño and La Niña conditions are also listed. Interestingly, there are more occurrences of type B when La Niña occurs, and, especially none but B for La Niña with EOF > 1.

**Table 1.** Statistics for the occurrence of EOF1 and the number of occurrences of each precipitation type. El Niño/La Niña information is appended.

|  |  |  |  | El Nino | La Nina |
|---|---|---|---|---|---|
| EOF1 > 1 |  | O: | 184 | 44 | 60 |
|  | EOF2+ | A: Mountain | 49 | 13 | 0 |
| | EOF2− | B: Plain | 53 | 12 | 30 |
| EOF1 >1 | EOF3+ | C: Seaward | 53 | 13 | 0 |
| | EOF3− | D: Inland | 45 | 14 | 0 |
| | EOF2+ | A: Mountain | 117 | 26 | 60 |
| | EOF2− | B: Plain | 155 | 32 | 120 |
| EOF1 < 1 | EOF3+ | C: Seaward | 123 | 29 | 0 |
| | EOF3− | D: Inland | 81 | 19 | 30 |

Figure 13 shows the average daily precipitation distribution for EOF1 > 1 overlaid with the average daily precipitation distribution for scores > 1 for each of A, B, C, and D. As divided in Table 1, each (A, B, C and D) average precipitation when EOF1 > 1 (EOF1 < 1) is shown in red (blue) contours. That is, the red (blue) in Figure 13a is the average of 49 (117) cases, and the red (blue) in (b) is the average of 53 (155) cases, etc. The precipitation composite peaks of EOFs 2 and 3 are at the same place with the eigenvector peaks. The peaks of A, C, and D exceed 20 mm/d (30 mm/d for D), but the center of B in EOF1 < 1 is shifted south. Average precipitation of type B maximum with EOF1 > 1 is 14 mm/day, but this is also more than EOF1 > 1. Thus, each peak of average precipitation has positive difference compared to EOF1 averaged precipitation.

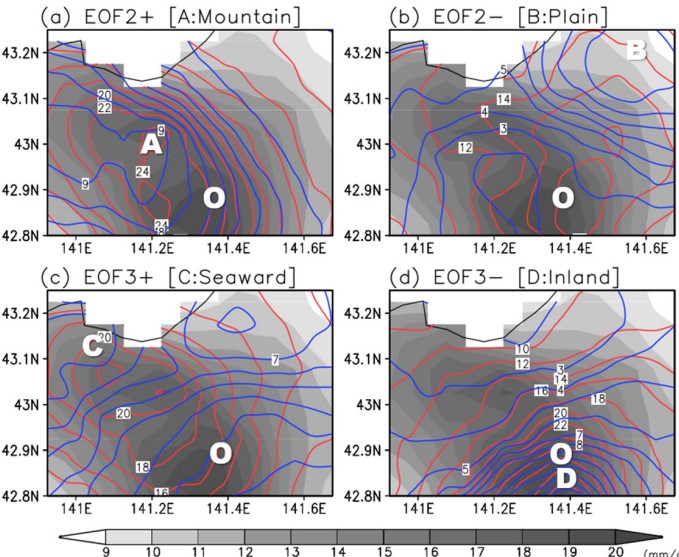

**Figure 13.** Average precipitation (grey shade) for EOF1 (score > 1) overlaid by red (blue) contours with average precipitation off each type (A, B, C, D) for EOF1 > 1 (EOF1 < 1).

*3.5. ENSO Precipitation Composite*

Then, we show the actual precipitation anomalies for the El Niño and La Niña winters. The precipitation anomalies for El Niño winters (i.e., 1998, 2003, 2010, and 2015) and those for La Niña winters (i.e., 1999, 2000, 2006, 2008, and 2011) are shown in Figure 14a,b. Distinct differences can be observed between the two maps. Generally, El Niño winter has less precipitation than average, but we found that El Niño years experienced more precipitation in the southern area of the target region, i.e., the inland (D) and mountains (A). La Niña years experienced more precipitation overall, and an especially large positive anomaly appears in the northeast (B), and a larger anomaly is observed around B and A, i.e., along the coast and in the plains.

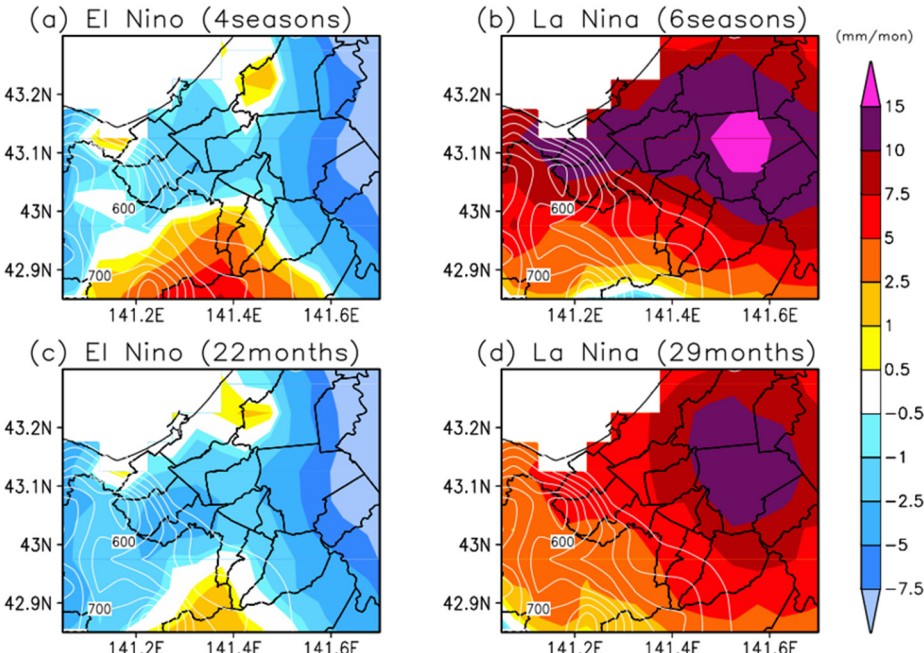

**Figure 14.** Seasonal and monthly precipitation composite anomalies. Color scale is the same for the four panels, and white contours indicate elevation. (**a**) El Niño winter (1998, 2003, 2010, and 2015) composite anomaly, and (**b**) La Niña winters (1996, 1999, 2000, 2006, 2008, and 2011). (**c**) El Niño winter monthly composite and (**d**) La Niña winter monthly composite.

For the 23 winters, the significance of the differences in the 4-case averaged El Niño to total variation and the same of 6-cases for La Niña are evaluated; however, they did not reach a significant difference at the 5% risk rate. As for each year anomaly, the high precipitation in 2015 (El Niño), 1996, 1999, and 2000 (La Niña) and the low precipitation to the south in 2008 (Lan Niña) showed a significant difference at the 5% risk rate.

Another attempt of ENSO precipitation composite is made for monthly precipitation. Based on monthly Nino-3 SST, we take the El Niño/La Niña anomaly (Figure 14c,d). The overall distribution pattern of Figure 14c,d is the same as in Figure 14a,b, but the mean anomaly is smaller than that of season, which may be due to the fact that it was evaluated from the running mean SST. Detailed analysis is a future issue, including the analysis physical linkage between the ENSO signal and local precipitation distribution.

## 4. Discussion

Based on the daily precipitation grid data over Sapporo, we clarified the relationship between regional winter precipitation distribution, dominant synoptic weather patterns, and their interannual variability. When the winter pattern is weak, precipitation tends to be brought to the mountain and inland regions of southwestern Sapporo (A and D). In contrast, when the winter pattern is strong, precipitation is more likely to occur in the seaward side and plain regions in the target area (B and C).

The purpose of this study was to look at the variability of horizontal precipitation distribution within the Sapporo area, but we also examined EOF1, which represents global precipitation. The high precipitation winter and frequency of occurrence of EOF1 > 1 did not agree with Farukh and Yamada [15]. In their 19 winter seasons, extreme snowfall days were found in 1999/2000, 2003/2004, and 1998/1999. Appearance of EOF1 > 1 was higher in all three years in our analysis, but top-3 was 2005, 1996, and 2013. This may be due to differences in the winter target months and in the data processing, even though the original MSS data handled were the same. They also indicate that the topmost one to six seasons appeared in the positive values of the Southern Oscillation Index (SOI). Although the indices used are different, this is consistent with our general observation in Figure 14 that there is more precipitation in La Niña and less precipitation in El Niño except in the southern part of the country. However, detailed comparisons will be made in the future, since the indices and months/years are somewhat different.

The occurrence of A, B, C, and D with respect to EOF1 > 1 and mean precipitation was also examined. It is found that type B (northeast plain precipitation) contributes more when EOF1 < 1 and areal precipitation is high. The type B is the one most likely to appear prominently in La Niña among the four types. As the result, we also showed more snow fell in the southwestern mountains and inland areas during El Niño winters (Figure 14b); on the contrary, we found that more snow fell in the northeastern plains and along the sea during La Niña winters (Figure 10). As for the difference in El Niño (4 years) or La Niña (6 years) average precipitation, seasonal precipitation anomaly was not significant. However, the appearance of the regional precipitation type seems to have physical meaning since their signal showed consistency with synoptic scale atmospheric circulation (SLP, 850 hPa wind). In addition, these results comply well with the fact that El Niño winters have weak winter patterns [4].

The relationship between the regional precipitation distribution and ENSO shown in Figures 9, 12 and 14 is very interesting: with the exception of 1998, types A,D appeared more frequently in El Niño, and precipitation was more in the mountains and inland areas of the Sapporo, and B,C appeared more frequently in La Niña and precipitation was more along the coast and in the plains. The winter period of 1997/1998 is known as a "super" El Niño [20]. This study has clarified that, with the exception of super-El Niño conditions, more precipitation falls in the mountain and inland regions during El Niño winters, and more precipitation is observed in the plains and seaward areas during La Niña periods. Note that the response of the Super El Niño to the winter monsoon in Japan is beyond the scope of this paper.

Regarding the relationship between winter pattern and Japanese winter snowfall, when a winter pattern is strong, i.e., high in the west and low in the east, more snow occurs in the mountains of Niigata Prefecture (approximately 500 km south of Sapporo) [5]. At first glance, Akiyama [5] and our study seem to contradict each other. However, this is due to the difference in the direction and position of the mountains relative to the winter monsoon between Niigata and Sapporo, and neither is incorrect. Analysis of western Hokkaido (Tachibana, 1995 [16]) indicated that, when a winter pattern is weak, the convergence of katabatic winds and northwestern monsoon causes more precipitation near the coastline. Our results and Tachibana (1995) also appear to be contradictory. However, this is due to the different spatial scales they are dealing with. Their "plain" includes our mountains, plains, and inland.

While ENSO and East Asian winter weather have been well studied (e.g. [17,18,20,21]), diagnostics of impacts to smaller regions like this are still scarce. Recently, fine grid precipitation data including mountain area are available, and it should be important to examine the relationship between precipitation distribution, climate index, and water vapor transport, etc. These efforts would contribute seasonal precipitation forecast and global warming impacts on water resources in the near future.

In terms of local effects of winter monsoon, the relationship between cold air mass outflow and ENSO [21] has been reported and can be studied in more detail. However, it

is recommended to use the indexes that are appropriate for the purpose together with an understanding of the physical processes mediated by the indexes, since the results are often not significant due to slight differences between the indexes used, such as Nino-3, Nino-3.4, and SOI. The correlation discussed in Section 3.3 (two variables shown in Figure 12) was not significant for the Southern Oscillation Index (SOI).

The ENSO composite is also made by using APHRO_JP (without MSS stations). It showed a similar pattern, but peak values are smaller even though we used the same years from the 23 winter seasons. Since APHRO_JP has more than 100 years, we can make a composite with more ENSO cases (not shown); however, the local difference that we showed in this study was difficult to quantify. This study shows that MSS is very effective in assessing the interannual variability signal of regional snowfall in Sapporo. Statistical analysis of 23 winters (1993–2015) of daily precipitation grid data reveals an ENSO-related [18] interannual signal. In fact, in February 2021, there was a heavy snowfall in Iwamizawa (northeast of Sapporo), and in January 2022, there was a heavy snowfall in Sapporo. Both of these events occurred under the La Niña conditions, which is consistent with the results shown in this study. Continued development and evaluation of daily grid precipitation that incorporate MSS are needed to predict the seasonal snowfall distribution and to mitigate the heavy snowfall disasters.

## 5. Conclusions

In order to understand how global signals, e.g., El Niño and La Niña, affect the regional distribution of snowfall in the Sapporo region, at first, we made a daily gridded precipitation dataset with APHRO_JP method [10] that accounts for orographic enhancement of precipitation by adding Sapporo Multi-Sensor network (MSS) data. Next, to analyze regional differences in winter daily precipitation, an EOF analysis is applied to the quantitative daily gridded data over Sapporo. We found that places showing large interannual variability are different from that of showing large daily precipitation variability. Eliminating an overall precipitation mode that is depicted by the EOF first component (EOF1), we defined four local precipitation types (A: Mountain, centered southwest mountain area; B: Plain, centered northeast; C: Seaward, coastal area; D: Inland, southern part) according to EOF2 and 3. Analysis of appearance of each type with atmospheric circulation clarified that types B and C tend to occur coherently in interannual winter fluctuations, and similarly, types A and D tend to occur coherently.

Furthermore, analyses of the occurrence of the precipitation events and associated atmospheric circulation patterns revealed that, with the exception of the Super El Niño winter of 1997/1998, types A and D appear more during El Niño winters that affect more snow fall in the southwestern mountains and inland areas. We also found that B and D appear more during La Niña winters, which brings more snow fall in the northeastern plains and along the coastal areas. Continued development and evaluation of daily grid precipitation that incorporate MSS are needed to adapt and to mitigate the heavy snowfall disasters and shortage of mountain snow water resources that happen in association with global climate signals like ENSO.

**Author Contributions:** Conceptualization, A.Y. and C.K.; methodology, A.Y. and C.K.; formal analysis, A.Y. and C.K.; investigation, A.Y. and C.K.; resources, A.Y.; data curation, A.Y. and C.K.; writing—A.Y. All authors have read and agreed to the published version of the manuscript.

**Funding:** This research was funded by the Environment Research and Technology Development Fund, 2-1602, Hirosaki University Institutional Research Grand, FY2019-2021, and the collaborative research program of the Disaster Prevention Research Institute of Kyoto University (2021G-12). Part of this work was also supported by the Grants-in-Aid for Scientific Research (16H02598, 18HP8021, 19HP8023, 21H02330).

**Institutional Review Board Statement:** Not applicable.

**Informed Consent Statement:** Not applicable.

**Data Availability Statement:** The APHRO_JP data is available at http://aphrodite.st.hirosaki-u.ac.jp/ (accessed on 1 March 2022). The APHRO_JP with Sapporo Multi-Sensor network (MSS) used in this study is available upon request.

**Acknowledgments:** The Sapporo Multi-Sensor Network data were provided by the Sapporo City Construction Bureau, Public Works Department, Snow Control Office, Project Division. We thank them for their cooperation.

**Conflicts of Interest:** The authors declare no conflict of interest.

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
