# Peer review of "Winter Orographic Precipitation and ENSO in Sapporo, Japan"

_atmosphere, doi:10.3390/atmos13091413_

Round 1

Reviewer 1 Report

---------- General comments ----------

The authors assessed the regional distribution of the snowfall in the Sapporo region of Japan in El Nino and La Nina conditions. It would be most useful to the local government and readers to have more information on snowfall pattern variability and its mitigation measures in the region. The Conclusions section of the article is found weak with a very short description of the whole paper.

EOF statistical technique has been used, however, the test of significance for such a small sample size i.e. El Nino are 4 events in total and La Nina are 6 events in total raises some doubts about the reliability of the results. 

---------- Specific issues (with page numbers where appropriate): ----------

1.     In the introduction section ……A similar kind of literature addressing the rainfall data in the mountainous region concerning climate change/variability will help in enriching the introduction section.

2.     It is understood that the geographical location of Japan is in the temperate region and therefore there is a very high probability of having snowfall as the type of precipitation. But it would be nice for some non-expert or general readers if some text elaborating on the snowfall as precipitation in winter is added.

3.     There is a problem with the term used to show the geographical location of Sapporo, it is better to use “northwest area” in place of “north-northwest area” (L 45).

4.     The sentence in L 45 “and previous studies have………………snow clouds.” looks out of context here and it is also without citation to any of the previous studies. Remove it.

5.     Named citation “Farukh and Yamada (2014)” must be used along with the citation required for the journal i.e. numerical for “[7]” (L 51).

6.     The use of “the pressure pattern of heavy snowfall” looks awkward here. What kind of pressure pattern is discussed here?

7.     The width of the legend in Figure 1 must be increased for the ease of colour gradient identification. The North direction mark is missing in the maps and the lat-long labels in the rightmost panel in Figure 1 is not visible. Write the unit in the scale bar of the map.

8.     Use subsequent levels of numerical bullet points for the sub-section (e.g. Precipitation data, Gridding method, EOF analysis, and Other data) in section 2.1 “Data used for analysis”.

9.     To remove the ambiguity in In L 77, keep “—more precipitation by the orographic lifting of moist air –” within the brackets.

10.  In L 84, replace “application” with “performing”.

11.  Citation of the data used and important literature must be there in section 2 of the article.

12.  Elaborate on the EOF analysis in section 2.

13.  Mention the name of the variables for which the data of ERA-interim reanalysis was used in the study and the data also need the proper citation.

14.  What is the logic behind using Nino3-SST, when many other similar indices are also available to represent the central pacific region?

15.  Contour labels in Figure 3 are so small to see.

16.  Correction in the legend of Figure 9 is needed, “(B+C)-(A+D)” must not be with line change style. The right side Y axis label should have the same orientation and therefore “Nino3” must follow it.

17.  Named citation must be used along with the citation required for the journal i.e. numerical for “[11]”, “[1]” (L 235, 28, ).

18.  Named citation must be used along with the citation required for the journal i.e. numerical for “[6]”, “[10]” (L 220,223,225,226)”.

19.  Revision in the caption of Figure 9 is needed “Numbers indicate year……….”.

20.  Figure 9 does not show the correlation (L239).

21.  How reliable are the results obtained in this study? The sample size is small i.e. El Nino are 4 events in total and La Nina are 6 events in total. Which significance test was used?

22.  Some more elaboration is needed in the Conclusions section of the article. It must be such that one should get the idea of What the whole research article is about and especially the results and their applicability must be highlighted in this section of the article.

Author Response

Reply to Reviewer 1フォームの始まり

Thank you very much for many constructive comments. My reply is written in blue colors.

The authors assessed the regional distribution of the snowfall in the Sapporo region of Japan in El Nino and La Nina conditions. It would be most useful to the local government and readers to have more information on snowfall pattern variability and its mitigation measures in the region. The Conclusions section of the article is found weak with a very short description of the whole paper.

I have rewritten that part and also abstract. Actually, another reviewer (reviewer-2) also pointed out the same thing (conclusions).

EOF statistical technique has been used, however, the test of significance for such a small sample size i.e. El Nino are 4 events in total and La Nina are 6 events in total raises some doubts about the reliability of the results. 

 This is a very critical matter, so that I carefully recalculated some statistics and made some new analyses. I tried to increase the number of ENSO events by using the long-term APHRO_JP. But these efforts made it clear that without additional MSS (Sapporo) data, precise precipitation pattern did not appear. Hence, I decided not to include the analysis result of using long-term data. Instead, I discussed about the importance of MSS data for adaptation/mitigation study. The result of this trial and error was, as a matter of fact, in the same direction as what you wrote in your last comment.

  Regarding test of anomaly difference of El Nino/La Nina pattern, significant level statistics was not obtained. We also discussed on this in the text.

 Another reviewer (reviewer-3) pointed out the relatively small variance of EOF2 and 3, and suggested to analyze EOF1 (85%). I carefully check EOF1 and compared the appearance (frequency) and precipitation amount and location for each mode. Interestingly, EOF1 (score > 1) sample was only 184 among 1789 samples, and EOF1 has large variability in the southwestern part of the target area (denoted “O”). On the other hand, SD of interannual variation show its peaks where EOF2 and 3 have large values.  Hence I consider that we can discuss the patterns of EOF2 and 3 as far as we can explain the physical correspondence with the atmospheric circulation and/or previous results.

Finally, I tone the ENSO composite results down and made precise explanation on this.

  1. In the introduction section ……A similar kind of literature addressing the rainfall data in the mountainous region concerning climate change/variability will help in enriching the introduction section.

Thank you very much. I really thought this comment deeply, and tried to search papers that refer APHRODIE reference papers (Yatagai et al. (2009, 2012)) because APHRODITE is used for mountainous region including Himalayas many times. However, some papers used this data inappropriately (i.e. without careful consideration of the number of gauges and/or change of input data), and others intended to use APHRODIE just for downscaling coarse climate models. Since this paper is not a review paper, we need not refer such studies comprehensively, even data other than APHRODITE.

 However, your suggestion to enrich the introduction is important in terms of addressing our purpose clearly. That means, development of precise long-term precipitation in the mountainous region is primarily important to promote this kind of study, and this point has been emphasized in my previous papers (Yatagai et al. 2005, 2012, 2019). Hence, I thoroughly rewrote the introduction and refer these papers. If you feel this as house-advertisement, I erase them.

It is understood that the geographical location of Japan is in the temperate region and therefore there is a very high probability of having snowfall as the type of precipitation. But it would be nice for some non-expert or general readers if some text elaborating on the snowfall as precipitation in winter is added.

Thank you very much. I added snowfall issue in the introduction and added a figure to show Japanese winter precipitation at Japan sea side (Fig.2a).

  1. There is a problem with the term used to show the geographical location of Sapporo, it is better to use “northwest area” in place of “north-northwest area” (L 45).

   Thank you very much for the notice. We replaced.

  1. The sentence in L 45 “and previous studies have………………snow clouds.” looks out of context here and it is also without citation to any of the previous studies. Remove it.

   Thank you very much for the comments. Actually, we wondered if we refer Asuma and Kikuchi (1987) which dealt with movements of radar echo in Sapporo area. However, it was written in Japanese and not an nternational journal paper. Now we removed the sentence according to your comment.

  1. Named citation “Farukh and Yamada (2014)” must be used along with the citation required for the journal i.e. numerical for “[7]” (L 51).

   Thank you very much for the comments. Although I am one of the co-editors of this special journal, I am not used to the reference style of this journal. In the original manuscript, the style “Tachibana (1995)[16]” seems not to be a problem, we chance “Farukh and Yamada (2014)” to “Farukh and Yamada (2014)[15]”.

  1. The use of “the pressure pattern of heavy snowfall” looks awkward here. What kind of pressure pattern is discussed here?

Not here. Farukh and Yamada (2014) discussed synoptic-scale atmospheric circulation patterns that are associated with heavy snowfall over Sapporo. They discussed SLP, 850 hPa moisture and wind, and 500 hPa geopotential height.

The previous explanation ( Farukh and Yamada (2014) discussed the pressure pattern of heavy snowfall in Sapporo on the scale of the Sapporo City.)  was awkward. We modified this sentence to explain “synoptic-scale atmospheric circulation patterns” for extremely heavy snowfall events.

  1. The width of the legend in Figure 1 must be increased for the ease of colour gradient identification. The North direction mark is missing in the maps and the lat-long labels in the rightmost panel in Figure 1 is not visible. Write the unit in the scale bar of the map.

We increased width of the legend in Fig.1, put larger font for lat-long labels, put the unit (km) beside the scale bar. However, we did not put “North direction mark”, because we clarified lat/lon and we put lines to show the noted lat/lon in the right panel. It clarifies North direction.

  1. Use subsequent levels of numerical bullet points for the sub-section (e.g. Precipitation data, Gridding method, EOF analysis, and Other data) in section 2.1 “Data used for analysis”.

Thank you very much for kind suggestion. We followed this journal’s style. However, we put subsequent levels of numerical bullet (2.1.1 – 2.1.4) according to your suggestion.

  1. To remove the ambiguity in In L 77, keep “—more precipitation by the orographic lifting of moist air –” within the brackets.

Thank you very much. We modified according to your suggestion.

  1. In L 84, replace “application” with “performing”.

Thank you very much. We modified according to your suggestion.

  1. Citation of the data used and important literature must be there in section 2 of the article.

      Some literatures are referred in section 1, even they are sometimes used as a data reference. However, we refer them again in section 2 according to your comment.

  1. Elaborate on the EOF analysis in section 2.

I am sorry but I do not fully understand your point. I merged the EOF results (the variance) from 3.1 (first paragraph) to section 2.3.1.

  1. Mention the name of the variables for which the data of ERA-interim reanalysis was used in the study and the data also need the proper citation.

    Mentioned the name of the variables. We have already referred Dee et al. (2011) as [8]. What else do we need to cite as a reference of the ERA-interim reanalysis?

  1. What is the logic behind using Nino3-SST, when many other similar indices are also available to represent the central pacific region?

As you kindly evaluated “It would be most useful to the local government and readers to have more information on snowfall pattern variability and its mitigation measures in the region.”, we would like to contribute local government and local people by this scientific result. In terms of this application, namely “local” response to ENSO, we, at first, take the index that Japan Meteorological Agency (JMA) uses to define El Nino/La Nina condition. We explained this in 2.1.4.

  1. Contour labels in Figure 3 are so small to see.

We repaired according to your comment.

  1. Correction in the legend of Figure 9 is needed, “(B+C)-(A+D)” must not be with line change style. The right side Y axis label should have the same orientation and therefore “Nino3” must follow it.

We repaired according to your comment.

  1. Named citation must be used along with the citation required for the journal i.e. numerical for “[11]”, “[1]” (L 235, 28, ).

We repaired according to your comment.

  1. Named citation must be used along with the citation required for the journal i.e. numerical for “[6]”, “[10]” (L 220,223,225,226)”.

We repaired according to your comment.

  1. Revision in the caption of Figure 9 is needed “Numbers indicate year……….”.

Thank you for the notification. Actually, we made a mistake to put Figure 6 caption on Figure 9 again. We wrote right caption. 

  1. Figure 9 does not show the correlation (L239).

Thank you for the notification. We refer results described in section 3.3.   

  1. How reliable are the results obtained in this study? The sample size is small i.e. El Nino are 4 events in total and La Nina are 6 events in total. Which significance test was used?

   This is a very important point. As for some winter season, absolute anomaly exceeds 2σ, and if normal distribution is assumed it exceeds 95 % confidence level. However, averaged anomaly of El Nino (4 events) and La Nina (6 events) did not reach. So, I honestly wrote this in the text and toned this result down.

  1. Some more elaboration is needed in the Conclusions section of the article. It must be such that one should get the idea of What the whole research article is about and especially the results and their applicability must be highlighted in this section of the article.

     With additional analyses for considering the major two points and your kind comments, I reached the same point what you commented. With 23 winter data that we can use MSS (even we update it until now, the number of events will not be drastically increased), it is difficult to get statistically significant level for your comment #21. We cannot go back for the historical period, but with this method (MSS + APHRODITE method) we obtained precise information of snow distribution that cannot be obtained by AMeDAS only. Continuous monitoring and update MSS analyses will be important for the application of the local government and inhabitants.  I wrote this point in “discussion” section.  Conclusion section and abstract were also rewritten.

Reviewer 2 Report

Introduction, specific information such as the literature review in view of the work undertaken is not discussed properly.

Also, the significance and scope of the work is not upto the mark and hence needs improvement.

the objectives should be defined clearly what planned to be done or included in the work.

results are presented in a lucid manner and however, lacks with proper justification and referencing.

related to figure 5 and 7, discussion needs improvement

The discussion on figure 10 in the case of seasonal wind composites for La Nino and El Nino studies is absurd and erroneous. The same discussion is not observed from the figure and discussion not supported strongly with any earlier works.

conclusions should be rewirtten based on the results and discussion.

Author Response

Reply comments to Reviewer 2

Thank you very much for many constructive comments. My reply is written in blue colors.

Introduction, specific information such as the literature review in view of the work undertaken is not discussed properly.

We added literature review.

Also, the significance and scope of the work is not upto the mark and hence needs improvement.

Actually, reviewer-1 gave us very constructive comments, and we emphasize the significance and scope of the work according to his/her comment. I was hesitated to refer papers written by myself. However, to make fine grid precipitation data with inputting additional data is one of the key issue to reach the objective of this work. Hence, I explained carefully the scope in addition to the general literature review.

the objectives should be defined clearly what planned to be done or included in the work.

Thank you for this comment. We clarified the purpose of this study at the last paragraph of section 1. Introduction.

results are presented in a lucid manner and however, lacks with proper justification and referencing.

I do not understand what you mean “lucid”, however, to understand each diagram and local pattern easily, I marked “O”, “A”, etc. For proper justification, reviewer-1 pointed statistical significance and reviewer-3 pointed the contribution of EOF1. Hence I added some new diagrams and added two subsections.

related to figure 5 and 7, discussion needs improvement

Figure 5 is now Figure 8, and Figure 7 is now Figure 10. Thank you very much for the comment. I added discussion based on the relationship between Figure 8 and 11. I added EOF1 statistics on Figure 10 and added discussion.

The discussion on figure 10 in the case of seasonal wind composites for La Nino and El Nino studies is absurd and erroneous. The same discussion is not observed from the figure and discussion not supported strongly with any earlier works.

Thank you for the comment. According to the three reviewer’s comments, we added figures and statistics. I am sorry but I do not understand what you mean “seasonal wind composites” here. After deep consideration of this comment and others, we put marks “O”, “A”, “B”, “C” and “D” to easily identify the place we discuss. I also added discussion of referring earlier works.

conclusions should be rewirtten based on the results and discussion.

Thank you for the comment. I really appreciate. Other reviewer (reviewer-1) also pointed out the same thing. I thoroughly rewrote conclusions, introduction (purpose of this study) and abstract.

Reviewer 3 Report

They authors explore the regional precipitation in Sapporo, Japan using daily gridded winter precipitation data. Their EOF results suggest, in general, more snow in the southwestern mountain and inland areas in the El Niño winters while in the northeastern plain and coastal areas in the La Niña winters during the past three decades.

This is an interesting study. I appreciate the analysis on the quantitative grid precipitation dataset based on the collection of rain-gauge data. However, I have a few questions that may merit authors’ attention.  

Though I agree with the authors that the EOF 2 and EOF3 may have physical meanings related to ENSO, but their contributions (7.6% and 4.1%) are fall smaller than EOF1 contribution (85%) to precipitation variance. So why not focus on the EOF1 that I guess may be related to global warming, which itself accounts for most precipitation variance?

Besides, I am a bit confused about Fig. 9 and its caption. Where are the blue and red cycles as mentioned in the caption?

Author Response

Reply to Reviewer-3フォームの始まり

 Thank you very much for many constructive comments. My reply is written in blue colors.

They authors explore the regional precipitation in Sapporo, Japan using daily gridded winter precipitation data. Their EOF results suggest, in general, more snow in the southwestern mountain and inland areas in the El Niño winters while in the northeastern plain and coastal areas in the La Niña winters during the past three decades.

This is an interesting study. I appreciate the analysis on the quantitative grid precipitation dataset based on the collection of rain-gauge data. However, I have a few questions that may merit authors’ attention.  

Though I agree with the authors that the EOF 2 and EOF3 may have physical meanings related to ENSO, but their contributions (7.6% and 4.1%) are fall smaller than EOF1 contribution (85%) to precipitation variance. So why not focus on the EOF1 that I guess may be related to global warming, which itself accounts for most precipitation variance?

I really appreciate your comments. To clarify this, I added analysis of EOF1 and compared the statistics and distribution with that of EOF2 and EOF3. The number of meaningful EOF1 (score >1) day is only 184 among 1789 sample days. The spatial pattern of standard deviation of interannual variability and that of daily variability are different. I added the following new figures/table according to your constructive comment.

 Figures 4a and b, 6, 13.

 Figure 10: added EOF1 statistics.

 Table 1

However, I have not investigated the EOF1 in terms of global warming, although I noticed that the precipitation of the first two winter is small so that we may show increasing trend.

Besides, I am a bit confused about Fig. 9 and its caption. Where are the blue and red cycles as mentioned in the caption?

Thank you very much for this notice. Actually, we made a mistake to write the caption of Fig.6 on Fig.9, too. We repaired.

Round 2

Reviewer 2 Report

Thank you for the authors in considering the reviewer comments on extensive manner and implemented well all suggestions and recommendations, made the revised paper improved strongly to meet the requirements of journal. Hence the present version of paper could be accepted for publication.